# Automatically Answering and Generating Machine Learning Final Exams

## Abstract

Can a machine learn machine learning? We propose to answer this question using the same criteria we use to answer a similar question: can a human learn machine learning? We automatically answer final exams in MIT's, Harvard's and Cornell's large machine learning courses and generate new questions at a human level. Recently, program synthesis and few-shot learning solved university-level problem set questions in mathematics and STEM courses at a human level. In this work, we solve questions from final exams that differ from problem sets in several ways: the questions are longer, have multiple parts, are more complicated, and span a broader set of topics. We provide a new dataset and benchmark of questions from machine learning final exams and code for automatically answering these questions and generating new questions. To make our dataset a reproducible benchmark, we use automatic checkers for multiple choice questions, questions with numeric answers, and questions with expression answers, and evaluate a large free language model, Meta's OPT, and compare the results with Open AI's GPT-3 and Codex. A student survey comparing the quality, appropriateness, and difficulty of machine-generated questions with human-written questions shows that across multiple aspects, machine-generated questions are indistinguishable from human-generated questions and are suitable for final exams. We perform ablation studies comparing zero-shot learning with few-shot learning, chain-of-thought prompting, GPT-3 and OPT pre-trained on text and Codex fine-tuned on code on a range of machine learning topics and find that few-shot learning methods perform best. We make our data and code publicly available for the machine learning community.

## 1 Introduction

Can a machine learn machine learning? This work presents a new dataset of machine learning final exams with 646 question parts and a benchmark of baselines using transformers and their respective grade performance, demonstrating that the best baseline performs at a human level. In university-level STEM courses, students complete assignments (including problem sets and labs) and exams throughout the course. Recent work has opened the door for a machine to solve course problem sets (Drori et al., 2022) using language models and few-shot learning. However, final exams remain challenging, and this work is the first to present a structured dataset of machine learning finals and a benchmark of baseline methods for answering them. Final exams differ from problem sets because they serve as a benchmark of cumulative understanding of material learned over a semester and evaluate the students' depth and breadth of expertise. Further, questions on final exams are longer, have multiple parts, span a broader set of topics, and are more complicated and nuanced.

All the above holds for MIT's and Cornell's Introduction to Machine Learning classes and Harvard's Machine Learning course. These are undergraduate courses with hundreds of students each semester, making them the largest undergraduate courses offered. Introduction to Machine Learning is a core class in the computer science program. The prerequisites for the course are Python Programming and Multivariate Calculus, with Introduction to Algorithms and Linear Algebra recommended. The class typically consists of weekly exercises, labs, quizzes, homework, a midterm, and a final exam. There were no final exams in at MIT for Fall 2020 and Spring 2020 due to COVID-19.

Introduction to Machine Learning final exams differ from problem sets in several ways, and the experience of solving each varies. First, finals are long, containing around nine questions with

around seven parts each. Final exam questions are also multifaceted and multi-stepped: different parts of a single question require applying different concepts and problem-solving skills, and parts may build upon each other. While weekly problem sets focus on a single topic, finals span topics from the entire semester. Further, final questions are often story-based problems that may require mathematical modeling. Due to the time constraint of these exams, finals are also designed to test core understanding and application of course material over rote calculations. Thus, asking a machine to answer questions from finals allows for testing whether the model is able to learn a breadth and depth of topics beyond problem sets.

In this work, we present a new dataset curated from final exams of MIT's and Cornell's Introduction to Machine Learning course and Harvard's Machine Learning class, totaling 646 question parts spanning the broad range of topics in machine learning: regression, classifiers, logistic regression, features, loss functions, neural networks, convolutional neural networks (CNNs), Markov decision processes (MDPs), recurrent neural networks (RNNs), reinforcement learning, clustering, decision trees, model selection, ensemble methods, Bayesian networks, hidden Markov models (HMMs), and optimization. Our dataset covers the finals given at MIT for semesters of Fall 2017, Spring 2018, Fall 2018, Spring 2019, Fall 2019, Spring 2021, Fall 2021, and Spring 2022, Harvard Spring 2015 and Spring 2021, and Cornell Spring 2017 and Fall 2018. Due to the COVID-19 pandemic, no finals were in the MIT course during 2020.

Table 1: MIT Spring 2021, Question 10.a: question part, solution, Codex Zero-Shot answer, program output, GPT-3 Zero-Shot answer, and GPT-3 Zero-Shot with CoT prompt answer.

| Topic | Classifiers |
|---|---|
| Question Part | Given a set of data $\mathcal{D}_{\text{train}} = \{(x^{(i)}, y^{(i)})\}$, a weighted nearest neighbor regressor has the form $$h(x; \theta) = \frac{\sum_{(x^{(i)}, y^{(i)}) \in \mathcal{D}_{\text{train}}} f(x, x^{(i)}; \theta) y^{(i)}}{\sum_{(x^{(i)}, y^{(i)}) \in \mathcal{D}_{\text{train}}} f(x, x^{(i)}; \theta)}.$$ A typical choice for $f$ is $$f(x, x'; \theta) = e^{-\theta \|x - x'\|^2}$$ where $\theta$ is a scalar and $\|x - x'\|^2 = \sum_{j=1}^{d}(x_j - x_j')^2$. Assume our training data $\mathcal{D}_{\text{train}} = ((1, 1), (2, 2), (3, 6))$. What is $h(10; 0)$? That is, letting $\theta = 0$, what is our prediction for $x = 10$? |
| Solution | 3 |
| Codex Zero-Shot Answer | ```python
import numpy as np

def f(x, x_i, theta):
  return np.exp(-theta * np.linalg.norm(x - x_i)**2)

def h(x, theta):
  D = np.array([(1, 1), (2, 2), (3, 6)])
  numerator = 0
  denominator = 0
  for x_i, y_i in D:
    numerator += f(x, x_i, theta) * y_i
    denominator += f(x, x_i, theta)
  return numerator / denominator

print(h(10, 0))
``` |
| Program Output | 3.0 |

| | |
|---|---|
| GPT-3 Zero-Shot Answer | $h(10;0) = \frac{f(10,1,0)y^{(1)}+f(10,2,0)y^{(2)}+f(10,3,0)y^{(3)}}{f(10,1,0)+f(10,2,0)+f(10,3,0)} = \frac{1\cdot1+1\cdot2+1\cdot6}{1+1+1} = \frac{9}{3} = 3$ |
| GPT-3 ZS with CoT Answer | $f(10,(1,1),0) = e^{-0\|10-(1,1)\|^2} = e^{-0\|9\|^2} = e^{-0.81} = 1$ 
 $f(10,(2,2),0) = e^{-0\|10-(2,2)\|^2} = e^{-0\|8\|^2} = e^{-0.64} = 1$ 
 $f(10,(3,6),0) = e^{-0\|10-(3,6)\|^2} = e^{-0\|7\|^2} = e^{-0.49} = 1$ 
 $h(10,0) = \frac{\sum_{(x^{(i)},y^{(i)})\in\mathcal{D}_{\text{train}}} f(x,x^{(i)},\theta)y^{(i)}}{\sum_{(x^{(i)},y^{(i)})\in\mathcal{D}_{\text{train}}} f(x,x^{(i)},\theta)} = \frac{1\cdot1+1\cdot2+1\cdot6}{1+1+1} = \frac{9}{3} = 3$ |

We verify that our results are not due to overfitting by including MIT finals from 2021 and 2022 that are unavailable online. Also, the MIT Spring 2022 final exam was given after GPT-3 and Codex were last updated, which means that the models were not trained on this data. The final exam questions have many parts, each posing a new problem, and each question in the dataset corresponds to one part. The questions in the finals are varied in topics and solution types. Solutions are primarily open-ended questions with some true/false and multiple-choice questions on theory, math, and code implementations.

We make the dataset publicly available and welcome others to use it to aid in developing and assessing new language models and methods. Due to the diversity of Intro to ML final questions, our dataset uniquely assesses advanced problem-solving and reasoning skills in machine learning, math, and natural language processing. This dataset opens the door to achieving breakthroughs in machine learning performance in machine learning final exams. In addition to the dataset, we present a benchmark using several baseline methods. We apply zero-shot and few-shot learning to GPT-3 and Codex, adding chain-of-thought prompting for GPT-3. We find that few-shot learning methods perform best. As shown in Table 2 the best performing methods pass the final exams, and their grade is comparable with human grades of MIT students on the same machine learning finals evaluated by the same human graders. We generate new final exam questions that are indistinguishable from human-written questions.

## 1.1 RELATED WORK

There is often thought that humans are generalists, whereas machines are specialists. However, large language models based on transformers such as GPT-3 (Brown et al., 2020), Gopher (Rae et al., 2021), and PaLM (Chowdhery et al., 2022), also called foundation models, are generalist learners. Specifically, in our setting, while humans care about the number of topics in an exam and therefore find finals more difficult than problem sets, foundation models effortlessly scale to many topics without re-training. Language models may be pre-trained on text and fine-tuned on specific datasets such as code, for example OpenAI's Codex (Chen et al., 2021), which allows generating programs from text. There are several ways to improve the mathematical reasoning ability of language models: (1) using chain-of-thought (CoT) prompting (Kojima et al., 2022; Wei et al., 2022), (2) using the top-k ranking solutions (Li et al., 2022) and merging them by voting (Wang et al., 2022) or least-to-most prompting (Zhou et al., 2022), and (3) using program synthesis and few-shot learning to generate code that answers questions (Drori et al., 2022).

| Grading: | Human | Human | Human | Machine |
|---|---|---|---|---|
| Answers: | Human | Human | Machine | Machine |
| Questions: | All | Non-Image | Non-Image | Non-Image Non-Open |
| MIT Spring 2021 | 75.84 | 80.77 | 62.09 | 64 |
| MIT Fall 2021 | 74.38 | 60.88 | 58.94 | 51.33 |
| MIT Spring 2022 | 69.07 | 70.82 | 68.86 | 73.53 |
| Mean | 73.10 | 70.82 | 63.29 | 62.95 |

Table 2: Human and machine grading of human and machine solved final exams. Mean human and machine grades on Introduction to Machine Learning final exams by semester. MIT Spring 2021, MIT Fall 2021, and MIT Spring 2022 final exams were unavailable online when GPT-3 and Codex were trained, ensuring that our results are not due to overfitting. Non-image grades consider question parts that do not contain images that are required for solving the question.

| Semester | Questions | Parts |
|---|---|---|
| MIT Fall 2017 | 10 | 61 |
| MIT Spring 2018 | 9 | 42 |
| MIT Fall 2018 | 10 | 60 |
| MIT Spring 2019 | 9 | 58 |
| MIT Fall 2019 | 8 | 61 |
| MIT Spring 2021 | 13 | 71 |
| MIT Fall 2021 | 8 | 86 |
| MIT Spring 2022 | 9 | 59 |
| Harvard Spring 2015 | 8 | 12 |
| Harvard Spring 2021 | 6 | 32 |
| Cornell Spring 2017 | 30 | 48 |
| Cornell Fall 2018 | 29 | 56 |
| Mean | 12.42 | 53.83 |
| Total | 149 | 646 |

| Topic | Questions | Parts |
|---|---|---|
| Regression | 10 | 62 |
| Classifiers | 24 | 85 |
| Logistic Regression | 3 | 10 |
| Features | 3.5 | 21 |
| Neural Networks | 19.5 | 87 |
| Loss Functions | 4 | 16 |
| CNNs | 9 | 65 |
| MDPs | 10 | 77 |
| RNNs | 7 | 33 |
| Reinforcement Learning | 11 | 60 |
| Clustering | 5 | 17 |
| Decision Trees | 14 | 48 |
| Model Selection | 5 | 16 |
| Ensemble Methods | 9 | 20 |
| Bayesian Networks | 1 | 6 |
| HMMs | 1 | 4 |
| Optimization | 10 | 16 |
| Bonus/Name | 3 | 3 |
| Mean | 8.28 | 35.89 |
| Total | 149 | 646 |

Table 3: The number of questions and parts in the final for each semester and topic of Introduction to Machine Learning. MIT Spring 2020 and Fall 2020 did not have final exams due to COVID-19. Topics can have half-questions attributed to them if a question has some parts under one topic and the other parts under another topic.

Much of the prior work focuses on high school or middle school level material (Qu et al., 2021). The first work to tackle university-level machine learning course problem set questions (Tran et al., 2021) used a transformer and GNN architecture and heavily relied on data augmentation. This resulted in overfitting and did not scale up to other types of questions or courses. Probability and statistics course problem-set questions have been answered (Tang et al., 2022) by probabilistic program synthesis with human performance. Problem-set questions from the core university math courses (Drori et al., 2022) have been automatically solved using few-shot learning and program synthesis at a human level. Other work considers university-level course questions across a variety of domains (Hendrycks et al., 2021) and identifying theorems (Srivastava et al., 2022). Prior work on question generation includes question–answer pair generation based on a text passage (Qu et al., 2021) and question text generation based on other questions (Drori et al., 2022).

## 2 DATASET

We present a new dataset of 646 question parts from a dozen recent final exams of MIT's and Cornell's Introduction to Machine Learning courses and Harvard's Machine Learning class. The dataset spans questions on the 17 machine learning topics covered in the courses: (1) regression, (2) classifiers, (3) logistic regression, (4) features, (5) loss functions, (6) neural networks, (7) convolutional neural networks (CNNs), (8) Markov decision processes (MDPs), (9) recurrent neural networks (RNNs), (10) reinforcement learning, (11) clustering, (12) decision trees, (13) model selection, (14) ensemble methods, (15) Bayesian networks, (16) hidden Markov models (HMMs), and (17) optimization. We make our data and code publicly available.[1]

The breakdown of questions, parts, points, and non-image points by each semester and topic are shown in Table 2. Each question in a final exam consists of multiple parts. Questions are written by providing set-up and context information first, followed by the question parts (which may come with additional information). Set-up and context information may contain (1) story elements (ex., character names, and motivations), (2) relevant definitions and equations, and (3) data points. We format questions in the dataset by concatenating the question context, any context or solutions from prior parts of the question required for answering the part, and the part's context and question. We split the questions into their corresponding parts. Questions consist of English text, mathematical notation, and images. Mathematical notation is represented in the dataset by LaTeX and images by screenshots from pdfs files. The types of question answers are diverse. A few are multiple-choice or

---

[1]Data and code are in the Supplementary Material.

true/false questions. Most are open-ended, for which the evaluation requires modeling the problem, mathematical manipulation, or code writing. Many questions require providing an explanation.

We used twelve final exams from different semesters for data curation. We had access to the Latex version for the three most recent semesters of MIT Spring 2021, Fall 2021, and Spring 2022, and therefore did not require transcription. For the nine remaining exams, MIT Fall 2017, Spring 2018, Fall 2018, Spring 2019, Fall 2019, Harvard Spring 2015 and Spring 2021, and Cornell Spring 2018 and Fall 2018, we had access to the pdf versions. In these cases, we used mathpix `mathpix.com` for an initial transcription, and curators then evaluated and manually corrected the input questions and verified the correctness of each input question.

We extract questions and solutions for all parts of all types of questions, including those that rely on images. We curated nine exams from publicly available pdf files. MIT Spring 2020 and Fall 2020 do not have final exams due to COVID-19. The three MIT exams between 2021 and 2022 were unavailable online; therefore, the model does not overfit their solutions. The aggregate average grades were available to the students and did not contain any personally identifiable information. Three duplicate questions were originally on the final exam of MIT Fall 2017 (questions 1, 3, 6) and appeared again in the final exam of MIT Spring 2022.

## 3 BENCHMARK

### 3.1 BASELINES

We provide a benchmark by comparing six baselines for answering the final exam questions: (1) GPT-3 with zero-shot learning, (2) GPT-3 with few-shot learning, (3) GPT-3 with zero-shot learning and chain-of-thought (CoT) prompting, (4) GPT-3 with few-shot learning and chain-of-thought (CoT) prompting, (5) Codex with zero-shot learning, and (6) Codex with few-shot learning.

Table 4 shows the prompt used for each approach. GPT-3 zero-shot uses the question as-is, whereas GPT-3 zero-shot with CoT uses the suffix "Let's think step by step." after the question to encourage multi-step output. Codex zero-shot uses the prefix "Write a program that answers" before the question within Python comments denoted by triple quotes """ to encourage Codex to write code. GPT-3 few-shot finds the closest questions in the embedding space, measured by cosine similarity, and uses them and their corresponding answers before the new question as examples in the prompt. Codex few-shot finds the closest questions in the embedding space also as measured by cosine similarity and uses these questions and their corresponding code as examples.

For students, a good study technique is to use previous final exams to review and practice for their upcoming final. We model this method by few-shot learning using the question–answer pairs (for GPT-3) or question–code (for Codex) with the closest question embeddings from previous finals. We implement this by considering all the exam questions, marking each question by its semester and year, and using only previous semesters' questions for few-shot learning. The MIT Fall 2017 and Spring 2022 exams contain three duplicate questions, and we handle these same questions the same way humans do by allowing few-shot learning in MIT Spring 2022 based on successful Fall 2017 zero-shot answers. It is reasonable that if a student studies all previous exams, there may be 8.5% of repeated question points. Since MIT Fall 2017, Harvard Spring 2015, and Cornell Spring 2017 are the first final exams in the corresponding universities, we do not perform few-shot learning on these.

### 3.1.1 COMPARISON WITH OPEN LANGUAGE MODELS

We also evaluated our dataset on an open-source language model, Meta's OPT-175B. OPT-175B is a model consisting of 175 billion parameters. Our dataset consists of final exam questions from machine learning courses and fit to be used by OPT-175B. Tables 5 and 6 compare the results of OpenAI GPT-3, OpenAI Codex, and Meta OPT. We evaluated OPT on only 163 question parts, since OPT was limited to handling questions under 256 characters in length. We implement the inference for the OPT-175B model using Alpa. Alpa is a particular framework designed for training and inference of large models. For the hardware, we use an 8x A100 PCIE cluster. The model requires about 560 GB of VRAM in our run case and each example takes nine minutes for inference.

| Method | Prompt |
|---|---|
| GPT-3 Zero-Shot | <question> |
| GPT-3 Few-Shot | Q: <similar question>
A: <similar question's answer>
Q: <question>
A: |
| GPT-3 Zero-Shot with CoT | Q: <question>
A: Let's think step by step. |
| GPT-3 Few-Shot with CoT | Q: <similar question>
A: <similar question's answer>
Q: <question>
A: Let's think step by step. |
| Codex Zero-Shot | """
Write a program that answers the following question: <question>
""" |
| Codex Few-Shot | """
Write a program that answers the following question: <similar question>
"""
<similar question's correct code>
"""
Write a program that answers the following question: <question>
""" |

Table 4: Input prompt for each of six baseline methods (1) GPT-3 Zero-Shot, (2) GPT-3 Few-Shot, (3) GPT-3 Zero-Shot with CoT, (4) GPT-3 Few-Shot with CoT, (5) Codex Zero-Shot, and (6) Codex Few-Shot. Similar questions (as measured by cosine similarity) are drawn from previous finals only.

## 3.2 GRADING

### 3.2.1 HUMAN GRADING

The questions are of different types: multiple-choice, numerical, expressions, and open-text. We grade answers and aim to keep all factors equal in grading human and machine answers. Human and machine answers are graded based on the number of points allocated to each question part, giving full, partial, or no credit for each answer. We approximate partial credit by assigning half-credit. The course staff graded student final exams, which included graduate TAs and instructors. Two of the same graduate TAs and the instructor that graded the student answers also graded the machine answers. Grading instructions are the same for grading student answers as grading machine answers.

### 3.2.2 AUTOMATIC GRADING

We label each question's answer type into one or two categories out of four options - multiple choice (MC), numerical, expression, or open. We consider answers multiple choice if the test-taker is presented with an enumerated list of choices, numerical if the answer is a number, expression if the answer includes variables or other notation, and open if the answer calls for free-response text. We categorize questions that have additional questions nested within them by the multiple relevant categories. Most often, this is the case when a question with one of MC, numerical, or expression, is followed by a follow-up question asking the student to explain their previous answer. The breakdown of the questions is: 98 are multiple-choice, 84 numerical, 81 expressions, and 204 are open. The 'Non-Open Points' column of Tables 7 and 8 show the answer type breakdown by number of points. Table 7 shows the number of question parts that do not rely on images, the number of points that do not rely on images, and the number of non-open question points in Introduction to Machine Learning finals for each semester. MIT Spring 2020 and Fall 2020 did not have final exams due to COVID-19. Table 8 shows the breakdown by topic. Our automatic grading uses string matching and regular expressions. In the case of multiple-choice results, we check that the output of the code is equal to the solution. In the case of numerical answers, we look for a matching integer or real number.

## 3.3 PERFORMANCE

Table 5 shows the machine grades by semester and Table 6 shows the machine grades by topic, excluding question parts that rely on images. We compare the average grade of GPT-3 with zero-shot

| Semester | GPT-3 ZS | GPT-3 FS | GPT-3 ZS CoT | GPT-3 FS CoT | Codex ZS | Codex FS | OPT ZS |
|---|---|---|---|---|---|---|---|
| MIT Fall 2017 | **38.21** | NA | 22.86 | NA | 21.43 | NA | NA |
| MIT Spring 2018 | 44.35 | 60.48 | 38.71 | **70.97** | 32.26 | 67.74 | 33.33 |
| MIT Fall 2018 | 51.99 | 52.18 | 61.63 | **64.17** | 49.78 | 54.00 | 47.54 |
| MIT Spring 2019 | 43.45 | 54.23 | 41.07 | **58.81** | 15.54 | 41.55 | 34.64 |
| MIT Fall 2019 | 54.92 | **77.05** | 29.92 | 58.20 | 26.23 | 61.48 | NA |
| MIT Spring 2021 | 44.33 | 55.81 | 53.45 | 60.21 | 33.62 | **62.09** | 33.77 |
| MIT Fall 2021 | 58.94 | **69.44** | 50.35 | 54.90 | 18.11 | 42.00 | 24.44 |
| MIT Spring 2022 | 42.78 | **68.86** | 32.03 | 53.48 | 51.01 | 65.46 | 60.71 |
| Harvard Spring 2015 | 85.71 | NA | 85.71 | NA | 50.00 | NA | 21.43 |
| Harvard Spring 2021 | 47.73 | **86.36** | 47.73 | 81.82 | 43.18 | **86.36** | 45.45 |
| Cornell Spring 2017 | 78.91 | NA | 80.86 | NA | 51.30 | NA | 21.88 |
| Cornell Fall 2018 | 36.45 | 53.27 | 44.39 | **61.21** | 42.52 | 56.07 | 28.97 |

Table 5: We benchmark different baselines for each semester, excluding question parts that rely on images. We compare the average grade of GPT-3 with zero-shot (ZS), GPT-3 with few-shot (FS) learning, GPT-3 with ZS, and chain-of-thought (CoT) prompting, GPT-3 with FS and CoT prompting, Codex with ZS, Codex with FS, and OPT with ZS. MIT Fall 2017, Cornell Spring 2017, and Harvard Spring 2015 were the first semester for each university, so few-shot learning results based on previous semesters are unavailable (NA). MIT Spring 2020 and MIT Fall 2020 did not have final exams due to COVID-19. MIT Spring 2021, MIT Fall 2021, and MIT Spring 2022 final exams were unavailable online when GPT-3 and Codex were trained, ensuring that the model is not overfitting. The result of the best-performing method for each semester is marked in bold.

| Topic | GPT-3 ZS | GPT-3 FS | GPT-3 ZS CoT | GPT-3 FS CoT | Codex ZS | Codex FS | OPT ZS |
|---|---|---|---|---|---|---|---|
| Regression | 31.71 | **50.00** | 25.61 | 40.85 | 40.24 | **50.00** | **50.00** |
| Classifiers | 38.18 | 46.21 | 26.28 | 42.35 | 18.88 | **53.74** | 50.00 |
| Logistic Reg. | 50.00 | 60.00 | **77.50** | **77.50** | 55.00 | 70.00 | 16.67 |
| Features | 58.65 | 75.96 | 53.85 | 77.31 | 68.85 | **81.54** | 10.00 |
| Loss Functions | NA | NA | NA | NA | NA | NA | NA |
| Neural Networks | 48.34 | 60.23 | 44.54 | **68.42** | 37.82 | 63.45 | 27.27 |
| CNNs | 37.50 | **53.58** | 28.36 | 47.81 | 13.38 | 36.77 | 23.83 |
| MDPs | 49.19 | 52.01 | 46.03 | **54.23** | 24.38 | 38.03 | 28.32 |
| RNNs | 61.46 | **71.88** | 57.29 | 66.14 | 12.50 | 40.63 | 39.28 |
| RL | 36.09 | 42.99 | 36.67 | **50.11** | 28.79 | 45.11 | 24.28 |
| Clustering | **100.00** | **100.00** | **100.00** | **100.00** | 50.00 | 50.00 | 63.33 |
| Decision Trees | 54.70 | **71.80** | 32.48 | 51.28 | 46.15 | 54.70 | 55.00 |
| Model Selection | 82.93 | 83.74 | 72.76 | **95.12** | 67.48 | 69.92 | 21.95 |
| Ensemble Methods | 27.89 | 50.00 | 22.12 | **66.35** | 32.69 | 50.00 | 13.46 |
| Bayesian Networks | **100.00** | **100.00** | **100.00** | **100.00** | 0.00 | 0.00 | **100.00** |
| HMMs | **100.00** | **100.00** | 50.00 | **100.00** | 100.00 | 100.00 | 100.00 |
| Optimization | 55.00 | 60.00 | 35.00 | 55.00 | 17.50 | **70.00** | 20.00 |

Table 6: We benchmark different baselines for each course topic, excluding question parts that rely on images. We compare the grade of GPT-3 with zero-shot (ZS), GPT-3 with few-shot (FS) learning, GPT-3 with zero-shot and chain-of-thought (CoT) prompting, GPT-3 with FS and CoT, Codex with zero-shot, Codex with few-shot learning, and OPT with ZS. The question parts on loss functions rely on image information and are therefore unavailable (marked NA). The result of the best-performing method for each semester is marked in bold.

(ZS), GPT-3 with ZS and chain-of-thought (CoT) prompting, GPT-3 with few-shot (FS) learning, GPT-3 with FS and CoT prompting, Codex with ZS, Codex with FS, and OPT with ZS. Fall 2017 is the first semester, so few-shot learning results based on previous semesters are unavailable (NA). Spring 2020 and Fall 2020 did not have final exams due to COVID-19. Spring 2021, Fall 2021, and Spring 2022 final exams were unavailable online when GPT-3 and Codex were trained, ensuring that the model is not overfitting content it has seen previously. The results consistently demonstrate that few-shot learning methods perform best across semesters and topics, as marked in bold.

## 3.4 LIMITATIONS

Our dataset consists of all question parts and their solutions, including images. However, our baseline methods do not handle questions that rely on an image containing the information required to solve the question since GPT-3 and Codex do not handle images. Tables 7 and 8 show the breakdown of the number of question parts and points of questions that do not rely on image information for answering the question. On average, 27.55% of the question parts, which make up 30.32% of the points in final

| Semester | Non-Image Parts / All | Non-Image Points | Non-Open Points |
|---|---|---|---|
| MIT Fall 2017 | 49 / 61 | 70 / 100 | 69 / 70 |
| MIT Spring 2018 | 27 / 42 | 62 / 100 | 59 / 62 |
| MIT Fall 2018 | 30 / 60 | 62 / 100 | 37.5 / 62 |
| MIT Spring 2019 | 41 / 58 | 70 / 100 | 48 / 70 |
| MIT Fall 2019 | 46 / 61 | 61 / 100 | 50 / 61 |
| MIT Spring 2021 | 51 / 71 | 62 / 100 | 43 / 61 |
| MIT Fall 2021 | 56 / 86 | 48 / 100 | 43 / 48 |
| MIT Spring 2022 | 46 / 59 | 68 / 100 | 53.5 / 68 |
| Harvard Spring 2015 | 8 / 12 | 70 / 90 | 35 / 70 |
| Harvard Spring 2021 | 15 / 32 | 22 / 53 | 11 / 22 |
| Cornell Spring 2017 | 48 / 48 | 128 / 128 | 74 / 128 |
| Cornell Fall 2018 | 51 / 56 | 107 / 120 | 71 / 107 |
| OPT Total | 418 / 585 | 759 / 1091 | 525 / 759 |
| Total | 468 / 646 | 830 / 1191 | 594 / 830 |

Table 7: The number of question parts that do not rely on images, the number of points that do not rely on images, and the number of non-open question points, in finals for each semester. MIT Spring 2020 and MIT Fall 2020 did not have final exams due to COVID-19.

| Topic | Non-Image Parts / All | Non-Image Points / All | Non-Open Points |
|---|---|---|---|
| Regression | 37 / 62 | 45 / 71 | 35.5 / 45 |
| Classifiers | 72 / 85 | 169 / 200 | 126.5 / 169 |
| Logistic Regression | 10 / 10 | 14 / 14 | 7 / 14 |
| Features | 18 / 21 | 26 / 38 | 22 / 26 |
| Loss Functions | 2 / 16 | 3 / 21 | 3 / 3 |
| Neural Networks | 66 / 87 | 112 / 153 | 100.5 / 112 |
| CNNs | 54 / 65 | 61 / 80 | 57 / 61 |
| MDPs | 38 / 77 | 39 / 121 | 32.5 / 39 |
| RNNs | 27 / 33 | 48 / 67 | 36 / 48 |
| Reinforcement Learning | 52 / 60 | 87 / 111 | 57 / 87 |
| Clustering | 5 / 17 | 30 / 55 | 15 / 30 |
| Decision Trees | 33 / 48 | 76 / 113 | 39 / 76 |
| Model Selection | 16 / 16 | 41 / 41 | 14 / 41 |
| Ensemble Methods | 20 / 20 | 52 / 52 | 31 / 52 |
| Bayesian Networks | 1 / 6 | 2 / 11 | 0 / 2 |
| HMMs | 1 / 4 | 1 / 7 | 0 / 1 |
| Optimization | 13 / 16 | 20 / 32 | 15 / 20 |
| Bonus/Name | 3 / 3 | 4 / 4 | 3 / 3 |
| Total | 468 / 646 | 830 / 1191 | 594 / 830 |

Table 8: The number of questions parts that do not rely on images, number of points that do not rely on images, and number of non-open question points in the finals for each topic of the course.

exams, are questions that rely on image information. The points attributed to the non-image parts are tallied, recorded, and used to calculate non-image percentage grades.

## 3.5 GENERATING NEW QUESTIONS

The creation of new, high-quality questions by course instructors and TA is often a time-consuming, high-effort process. These new questions must be varied from past questions while still testing the same core concepts. We explore the potential of using GPT-3 to write exam content efficiently by using the dataset of exam questions to generate new questions automatically. We use questions from our dataset as prompts to create new high-quality questions not present in our dataset. We create a list of various questions in our curated dataset and use the resulting list to prompt GPT-3 to create a new question. The supplementary material demonstrates the results of this process for each topic in the course. The Appendix consists of new generated questions and the closest question from our dataset as measured by the cosine similarity of the embedding of each question. These new questions are diverse and qualitatively similar to questions on previous MIT final exams. This provides an efficient way for course TAs and instructors to generate new final questions.

### 3.5.1 STUDENT SURVEY

To evaluate the machine-generated questions, we conducted an anonymous online student survey comparing them with the human-written questions in terms of quality, appropriateness relative to

the course, and question difficulty. We surveyed 15 students who have taken the Introduction to Machine Learning course or its equivalent. The survey was optional and included informed consent, with the following description: "We are conducting a survey to assess the quality and difficulty of automatically generated questions for an introductory machine learning course final. You will be presented with a series of questions, either human-written (taken from an actual course final exam) or machine generated, but you will not be told the source of a given question. For each question, you will be asked (a) whether you think the question is human-written or machine-generated, (b) whether the question is appropriate for the given course final, and finally (c) how you would rate the difficulty of the question. Please carefully read each question and answer to the best of your ability".

We randomly sampled one generated question and its closest (measured by cosine similarity) original, human-written question for each of the twelve machine learning topics. Students were asked to read these 24 questions in the survey, mixed and presented randomly, and then answer three questions for each: (1) "Is the question human-written or machine-generated?", (2) "Is the question appropriate or not appropriate for the specific course final?", and (3) "What is the question's difficulty level on a scale between 1 (easiest) and 5 (hardest)?". We ask the students to provide ratings and not to solve the questions. The results of our survey are as follows: Out of the human-written questions, students identified 56.11% of them correctly as human-written and 43.89% incorrectly as machine-generated. Of the machine-generated questions, students identified 45% of them correctly as machine-generated and 55% of them incorrectly as human-written. The difficulty ratings were between 1 (the easiest) and 5 (the hardest). Students rated machine-generated questions with a difficulty level of 2.55 with a 1.11 standard deviation and rated human-written questions with a difficulty level of 2.85 with a 1.12 standard deviation. Students rated machine-generated questions as appropriate 82.6% of the time and human-written questions as appropriate 85.0% of the time.

The conclusions we draw from the survey are that (1) survey participants considered human-written questions to be as likely to be human-written or machine-generated, and similarly, machine-generated questions were considered equally likely to be machine-generated as human-written, (2) survey participants considered the machine-generated questions slightly easier than human-written questions, and (3) survey participants considered machine-generated questions as appropriate as human-written questions. Based on these results, we conclude that across multiple aspects, the machine-generated questions are highly similar to human-generated questions and can be adapted to generate questions for machine learning courses.

### 3.6 IMPLEMENTATION DETAILS

We use the latest OpenAI GPT-3 and Codex models and do not re-train these very large language models. We fix all the hyperparameters of the models so that the answers are deterministic and reproducible. Specifically, we set both top P, which controls diversity, and sampling temperature, which controls randomness, to 0. The frequency and presence penalties are also set to 0, and we do not halt on any stop sequences. We allow diversity for generating new questions by setting the top P and temperature to 0.1. We run Codex with an upper bound of generating programs with 1024 tokens. We use the OpenAI text-davinci-002 and code-davinci-002 engines for generating text and programs. For few-shot-learning and question generation, we use the text-similarity-babbage-001 engine to embed the questions and find the closest questions in the dataset by cosine similarity. The running time for answering or generating each question part is a few seconds.

## 4 CONCLUSIONS

We present a dataset and benchmark for answering and generating university-level final exams in machine learning. Machine performance and human performance are evaluated by the same graders and grading instructions, as well as by automatic checkers. A comparison of baselines shows that few-shot learning methods perform best across semesters and topics. A limitation of our work is that our benchmark does not consider questions that rely on images for their solution. This work may result in improving students learning for final exams, help course staff generate questions for finals, and compare levels of difficulty of exams across semesters and schools. This work includes final exams from MIT's and Cornell's Introduction to Machine Learning classes and Harvard's Machine Learning course. We hope this dataset and benchmark serve the machine learning community and advance the state-of-the-art in the field.

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

# A  APPENDIX

Table 9: Generating new questions: example of a new question for each topic automatically generated and the closest question in the dataset based on cosine similarity of the questions embeddings.

| Topic | Question | Similarity |
|---|---|---|
| Regression | Generated Question: "We're given a data set $D = \left\{ \left( x^{(i)}, y^{(i)} \right) \right\}_{i=1}^{n}$, where $x^{(i)} \in R^d$ and $y^{(i)} \in R$. Let $X$ be a $d \times n$ matrix in which the $x^{(i)}$ are the columns and let $Y$ be a $1 \times n$ vector containing the values of $y^{(i)}$. Using the ordinary least-squares formula, we can compute $W_{ols} = \left( XX^T \right)^{-1} XY^T$ Using ridge regression, we can compute $W_{\text{ridge}} = \left( XX^T + \lambda I \right)^{-1} XY^T$ We decide to try to use these methods to initialize a single-unit neural network with a linear activation function. Assume that $XX^T$ is neither singular nor equal to the identity matrix, and that neither $W_{\text{ols}}$ nor $W_{\text{ridge}}$ is equal to $(0, 0, \ldots, 0)$. Consider a neuron initialized with $W_{\text{ridge}}$. Provide an objective function $J(W)$ that depends on the data, such that batch gradient descent to minimize $J$ will have no effect on the weights, or argue that one does not exist." | 0.945 |
| | Closest Question: "We're given a data set $D = \left\{ \left( x^{(i)}, y^{(i)} \right) \right\}_{i=1}^{n}$, where $x^{(i)} \in R^d$ and $y^{(i)} \in R$. Let $X$ be a $d \times n$ matrix in which the $x^{(i)}$ are the columns and let $Y$ be a $1 \times n$ vector containing the values of $y^{(i)}$. Using the analytical regression (ordinary least-squares) formula, we can compute $$W_{ols} = \left( XX^T \right)^{-1} XY^T$$ Using ridge regression, we can compute $W_{\text{ridge}} = \left( XX^T + \lambda I \right)^{-1} XY^T$ We decide to try to use these methods to initialize a single-unit "neural network" with a linear activation function and no offset: $h(x; W) = W^T x$. Assume that $XX^T$ is invertible and not equal to the identity matrix, and that neither $W_{ols}$ nor $W_{\text{ridge}}$ is equal to $(0, 0, \ldots, 0)$. Note also that we are not using an explicit offset/bias term. Rory has solved many problems from this particular domain before and the solution has typically been close to $W^* = (1, \ldots, 1)^T$. Define an objective function $J(W)$ that we could minimize in order to obtain good estimates for Rory's next problem, even with very little data." | |
| Classifiers | Generated Question: "Consider a binary classification problem with two classes, +1 and -1. Assume that the training data is linearly separable. Consider the following two models:
Model 1: $g(x) = \text{sgn}(w^T x)$
Model 2: $g(x) = \text{sgn}(w^T x + b)$
Assume that the training data is linearly separable. Which model is more likely to overfit? | 0.782 |
| | Closest Question: "In some cases, we will have a validation set in addition to training and test sets. Assume the validation set is approximately the same size as the test set. This validation set is often used to tune hyperparameters such as $\lambda$. Imagine we have trained a classifier using regularization, with $\lambda$ chosen based on performance on the training set. Which will have the highest accuracy the training set, the validation set or the test set?" | |

Table 9: Generating new questions: example of a new question for each topic automatically generated and the closest question in the dataset based on cosine similarity of the questions embeddings.

| Topic | Question | Similarity |
|---|---|---|
| Logistic Regression | Generated Question: "In logistic regression, we use the sigmoid function to "squish" the output of our linear function to be between 0 and 1. What is the output of the sigmoid function when the input is 0?" | 0.810 |
| | Closest Question: "Consider the simplest of all neural networks, consisting of a single unit with a sigmoid activation function: $h(x; w = \sigma(w_0 + w_1 x)$ where $\sigma(z) = (1 + exp(-z))^{-1}$ Let's start with a classifier defined by $w_0 = -1$ and $w_1 = 1$. Which range of input values x are classified as positive? Which as negative?" | |
| Features | Generated Question: "You are working on a project to predict how successful a given Kickstarter campaign will be, based on information about the campaign (such as the amount of money pledged, the number of backers, the category of the project, the country where the project is based, and so forth). One of the features you want to use is the project's title. What is a good way to encode the project's title for use in a machine-learning predictor?" | 0.761 |
| | Closest Question: "Mac O'Larnin is considering selling an app on Frugal Play. You have a friend with inside info at Frugal, and they're able to share data on how previous apps have performed on the store. Mac decides that he will learn a neural network with no hidden layer (i.e., consisting only of the output layer). He needs help in figuring out the precise formulation for machine learning. What is the best way to encode the app characteristic 'Suitable for people ages (2–4, 5–10, 11–15, 16 and over)' as a feature for an input to the neural network? Choose from among the following: multiple unary features (one-hot encoding), multiple binary features (thermometer encoding), an integer or real-valued feature. Also give the exact function that maps each input to its corresponding feature(s)." | |

Table 9: Generating new questions: example of a new question for each topic automatically generated and the closest question in the dataset based on cosine similarity of the questions embeddings.

| Topic | Question | Similarity |
|---|---|---|
| Loss Functions | Generated Question: "We have a data set of images, each of which is either a picture of a dog or a picture of a cat. We want to train a binary classifier to distinguish between the two classes. Which of the following is a better choice of loss function for this task: (a) The 0-1 loss, which gives a loss of 1 if the classifier predicts the wrong class and a loss of 0 otherwise. (b) The cross-entropy loss, which is given by -log(p) if the correct class is predicted and -log(1-p) otherwise, where p is the predicted probability of the correct class. (c) The hinge loss, which is given by max(0, 1-s) if the correct class is predicted and max(0, 1+s) otherwise, where s is the predicted score for the correct class." | 0.790 |
| | Closest Question: "Consider the following 2D dataset in (x,y) format: ((1,-1), +1), ((1,1), +1), ((1,2.5),+1), ((2,-2),-1), ((2,1),+1),((2,3),+1),((5,-1),-1),((5,-2),-1). Break ties in distance by choosing the point with smaller $x_1$ coordinate, and if still tied, by smaller $x_2$ coordinate. Compute the leave-one-out cross validation accuracy of the 3-nearest-neighbor learning algorithm on this dataset." | |
| Neural Networks | Generated Question: "Consider a neural network with two hidden layers, each with two units. The input layer has two units, and the output layer has one unit. The hidden layers use the sigmoid activation function, and the output layer uses the linear activation function. The weights from the input layer to the first hidden layer are w11 = 1, w12 = 1, w21 = 1, and w22 = 1. The weights from the first hidden layer to the second hidden layer are w11 = 1, w12 = 1, w21 = 1, and w22 = 1. The weights from the second hidden layer to the output layer are w11 = 1, w21 = 1. The bias terms are all zero. What is the output of the neural network for the input x1 = 1, x2 = 1?" | 0.880 |
| | Closest Question: "A neural network is given as $Z^1 = X * W^1$, $A^1 = f1(Z^1)$, $Z^2 = W^2 * A^1$, $\hat{y} = f^2(Z^2)$. Specifically, the input $X$ is a $4 \times 1$ column vector, $\hat{y}$ is a $1 \times 1$ scalar. $W^2$ is a $3 \times 1$ vector. We also know that, $Z^1 = (W^1)^T X$ and $Z^2 = (W^2)^T A^1$. What are the dimensions of $Z^2$?" | |
| CNNs | Generated Question: "Suppose we have a 3x3 image and we use a 2x2 filter with stride 1. What are the dimensions of the output image?" | 0.895 |
| | Closest Question: "A neural network is given as $Z^1 = X * W^1$, $A^1 = f^1(Z^1)$, $Z^2 = W^2 * A^1$, $\hat{y} = f^2(Z^2)$. There is only one data point which is: $X = [1, 1, 1, 1]^T$ and $y = [1]$. If $W^1$ and $W^2$ are both matrices/vectors of all ones, what is the resulting Loss where the Loss = $(y - \hat{y})^2$?" | |

Table 9: Generating new questions: example of a new question for each topic automatically generated and the closest question in the dataset based on cosine similarity of the questions embeddings.

| Topic | Question | Similarity |
|---|---|---|
| MDPs | Generated Question: "Consider an MDP with four states, called $A, B, C,$ and $D$, and with two actions called Move and Stay. The discount factor $\gamma = 0.9$. Here is a reminder of the Q-learning update formula, based on experience tuple $(s, a, r, s')$ : $$Q(s, a) := (1 - \alpha)Q(s, a) + \alpha \left( r + \gamma \max_{a'} Q\left(s', a'\right) \right)$$ Let $\alpha = 1$. Assume we see the following state-action-reward sequence: A, Move, 0 B, Move, 0 C, Move, 1 A, Move, 0 B, Move, 0 C, Move, 1 With Q-values all starting at 0, we run the Q-learning algorithm on that state-action sequence. Provide the q-learning value for Q(C, Move)." Closest Question: "Consider an MDP with four states, called $A, B, C,$ and $D$, and with two actions called Move and Stay. The discount factor $\gamma = 0.9$. Here is a reminder of the Q-learning update formula, based on experience tuple $(s, a, r, s')$ : $$Q(s, a) := (1 - \alpha)Q(s, a) + \alpha \left( r + \gamma \max_{a'} Q\left(s', a'\right) \right)$$ Let $\alpha = 1$. Assume we see the following state-action-reward sequence: A, Move, 0 B, Move, 0 C, Move, 1 A, Move, 0 With Q-values all starting at 0, we run the Q-learning algorithm on that state-action sequence. Provide the q-learning value for Q(A, move)." | 0.988 |
| RNNs | Generated Question: "Consider the following RNN: $s_t = \tanh(w_1 x_t + w_2 s_{t-1} + b)$ , $y_t = w_3 s_t + b_2$ . Assume $s_0 = 0$ and $b_2 = 0$. What values of $w_1$, $w_2$, $w_3$ and $b$ would generate output sequence $[0, 0, 0, 1, 1, 1, 1]$ given input sequence $[0, 0, 0, 1, 0, 1, 0]$" Closest Question: "Ronnie makes a simple RNN with state dimension 1 and a *step* function for $f_1$, so that $s_t = \text{step}(w_1 x_t + w_2 s_{t-1} + b)$ where $\text{step}(z) = 1$ if $z > 0.0$ and equals 0 otherwise, and where the output $$y_t = s_t \ .$$ Assuming $s_0 = 1$, we want to generate output sequence $[0, 0, 0, 1, 1, 1, 1]$ given input sequence $[0, 0, 0, 1, 0, 1, 0]$. Rennie thinks this is not possible using Ronnie's architecture. Rennie makes an argument based on the relationships in the table above. Is Rennie right?" | 0.907 |
| Reinforcement Learning | Generated Question: "What is the tabular Q-learning update equation, based on experience tuple $(s, a, r, s')$?" Closest Question: "b is the bias for classifier network. What are dimensions of b for multi k-class classification?" | 0.796 |

Table 9: Generating new questions: example of a new question for each topic automatically generated and the closest question in the dataset based on cosine similarity of the questions embeddings.

| Topic | Question | Similarity |
|---|---|---|
| Clustering | Generated Question: "Suppose that we have a dataset with $n$ data points, $k$ clusters, and $d$ features. After running the $k$-means algorithm, the within-cluster sum of squared errors (WCSS) is given by: $$\frac{1}{n} \sum_{i=1}^{n} \|\mathbf{x}_i - \mu_{y_i}\|^2$$ where $y_i$ is the cluster label of the $i$th data point, and $\mu_{y_i}$ is the cluster center associated with the $i$th data point. The within-cluster sum of squared errors (WCSS) is a measure of how well the clusters fit the data. Suppose that we have two datasets, $X_1$ and $X_2$, where $X_1$ has $n_1$ data points and $X_2$ has $n_2$ data points. We run the $k$-means algorithm on both datasets. We find that the WCSS for $X_1$ is smaller than the WCSS for $X_2$. Does this imply that the clusters for $X_1$ are better than the clusters for $X_2$? Why or why not?" 

 Closest Question: "Consider the following 2D dataset in (x,y) format: ((1,-1), +1), ((1,1), +1), ((1,2.5),+1), ((2,-2),-1), ((2,1),+1),((2,3),+1),((5,-1),-1),((5,-2),-1). We will construct a tree using a greedy algorithm that recursively minimizes weighted average entropy. Recall that the weighted average entropy of a split into subsets A and B is: (fraction of points in $A$) $\cdot$ $H\left(R_{j,s}^A\right)$ + ( fraction of points in $B$) $\cdot$ $H\left(R_{j,s}^B\right)$ where the entropy $H\left(R_m\right)$ of data in a region $R_m$ is given by $H\left(R_m\right) = -\sum_k \hat{P}_{mk} \log_2 \hat{P}_{mk}$. The $\hat{P}_{mk}$ is the empirical probability, which is in this case the fraction of items in region $m$ that are of class $k$. Some facts that might be useful to you: H(0) = 0, H(3/5) = 0.97, H(3/8) = 0.95, H(3/4) = 0.81, H(5/6) = 0.65, H(1) = 0. Draw the decision tree that would be constructed by our tree algorithm for this dataset. Clearly label the test in each node, which case (yes or no) each branch corresponds to, and the prediction that will be made at each leaf. Assume there is no pruning and that the algorithm runs until each leaf has only members of a single class." | 0.767 |
| Decision Trees | Generated Question: "The Gini score is a measure of how often a randomly chosen element would be incorrectly labeled if it was randomly labeled according to the distribution of labels in the subset. It ranges from 0 to 1, with 0 meaning that there are no mislabeled elements and 1 meaning that the subset is perfectly mixed. Explain whether or not it would be a good idea to use the Gini score as a scoring function for pruning decision trees." 

 Closest Question: "There are different strategies for pruning decision trees. We assume that we grow a decision tree until there is one or a small number of elements in each leaf. Then, we prune by deleting individual leaves of the tree until the score of the tree starts to get worse. The question is how to score each possible pruning of the tree. Here is a definition of the score: The score is the percentage correct of the tree on a separate validation set. Explain whether or not it would be a good idea and give a reason why or why not." | 0.867 |

