# OpenReview forum: "Automatically Answering and Generating Machine Learning Final Exams"
_ICLR.cc/2023/Conference — Submitted to ICLR 2023_

### Official Review · Reviewer_bhJL · 2022-10-19

**Confidence:** 4
**Correctness:** 3
**Technical Novelty And Significance:** 3
**Empirical Novelty And Significance:** 3
**Recommendation:** 8

**Clarity, Quality, Novelty And Reproducibility:**


This work is of high quality since it presents a novel approach by being the first study to curate a machine learning dataset of final exams. This work is also reproducible since the authors made their data and code publicly available for the machine learning community.

**Details Of Ethics Concerns:**

None.

**Strength And Weaknesses:**

Strength: This is the first study to present a dataset of machine learning final exams
Weakness 1: A lot of details are missing with regards to how the dataset was curated. An illustration/figure to show the method used in this work would be advantageous.
 Weakness 2: The recommended models do not consider questions that rely on images for their automated solution.

**Summary Of The Paper:**

This work presented a new dataset that contains final exam questions, and compares the performance of popular transformer models for answering them. The dataset includes questions that arise from a wide range of topics in machine learning including loss functions, RNNs, reinforcement learning, and etc.

**Summary Of The Review:**

This paper presents an appealing and novel approach by generating a new dataset of machine learning final exams. This work also provided a benchmark of models for answering the final exam questions.

---

### Official Review · Reviewer_FkVg · 2022-10-25

**Confidence:** 4
**Correctness:** 4
**Technical Novelty And Significance:** 4
**Empirical Novelty And Significance:** 4
**Recommendation:** 5

**Clarity, Quality, Novelty And Reproducibility:**

The work is clear, well evaluated, novel and reproducible with the supplemental material.

**Strength And Weaknesses:**

Strengths:
- Much needed dataset for the community and interesting topic
- Strong GPT- and Codex-based baselines, as well as baselines with Open LMs
- Extensive experiments provide benchmarking results to start off of
- Great experiments and promising results with generated questions

Weaknesses:
- No major weakness

Notes:
- The references should be in parentheses
- In Table 5, there is the exact same result between GPT3-ZS and GPT-3 FS in Fall 2021, is that a typo?
- "We extended this work to include final exams from Harvard’s and Cornell’s machine learning classes as well. " Are these questions included or to be included in the future?

**Summary Of The Paper:**

The authors introduce a new university exam dataset spanning questions on 12 machine learning topics covered at MIT's Introduction to Machine Learning course. This joins a thin body of work of exam datasets that have mostly focused on high school and lower levels. The dataset and code are open-sourced and uploaded as part of the Supplemental Material. The exams span multiple quarters of this class. The experiments include benchmarking LMs on this dataset, as well as generating new exam questions. The results show that the generated questions are of equal quality as the human-written ones, have the same average difficulty rating, and get recognized as human-written with the same probability.

UPDATE: After discussion with my fellow reviewers and area chair, I agree that there are valid concerns about the size of the dataset. While we all praise the authors for the originality of the work and consider it valuable to the community, the dataset as presented may be too small for the community to use. We recommend trying to expand the dataset and resubmitting the paper.

**Summary Of The Review:**

This work introduces a novel dataset of university level machine learning exam questions and answers. This is a much needed resource for the community, and the results of the benchmarking is valuable, as well as the ones for exam question generation.

---

### Official Review · Reviewer_ffgb · 2022-10-25

**Confidence:** 4
**Clarity, Quality, Novelty And Reproducibility:** What's the difference between the pro…
**Correctness:** 3
**Technical Novelty And Significance:** 2
**Empirical Novelty And Significance:** 2
**Recommendation:** 3

**Strength And Weaknesses:**

Strength:
1. A quite interesting dataset
2. Solid baselines

Weakness:
1. Seems like one more dataset from BIG-bench, https://github.com/google/BIG-bench . Not too much novelty
2. The paper is not well-written. Not sure why Figure 1 is shown in the paper. It only shows students taking the final exam which is not related to the dataset.

**Summary Of The Paper:**

The authors provide a new dataset and benchmark of questions from machine learning final exams and code for automatically answering these questions and generating new questions. They build baselines by applying zero-shot and few-shot learning to GPT-3 and Codex, adding chain-of-thought prompting for GPT-3.

**Summary Of The Review:**

I think this is an interesting dataset, but it is like one dataset from BIG-bench. I think we don't need every dataset from BIG-bench to be published.

---

### Author Response · Authors · 2022-11-17
**Response to reviewers' comments**

We thank the reviewers for their time and have addressed all the comments.

Reviewer ffgb
1. Reviewer's comment: "Seems like one more dataset from BIG-bench, https://github.com/google/BIG-bench . Not too much novelty"
Response: BIG-bench is based on the paper: https://arxiv.org/pdf/2206.04615.pdf by Srivastava, Aarohi, et al. "Beyond the Imitation Game: Quantifying and extrapolating the capabilities of language models." arXiv preprint arXiv:2206.04615 (2022).
BIG-bench consists of a broad range of task topics, including linguistics, childhood development, math, common-sense reasoning, biology, physics, social bias, and software development; however, it does not consist of university-level final exams. Our work is vastly different from the BIG-bench dataset. We curate a challenging dataset of questions from university machine learning final exams in top universities' machine learning classes, specifically at MIT, Harvard, and Cornell - see the updated version.
2. Reviewer's comment: "Not sure why Figure 1 is shown in the paper. It only shows students taking the final exam which is not related to the dataset."
Response: We removed figure 1 and improved the paper's readability.

Reviewer FkVg
1. Reviewers comment: The references should be in parentheses
Response: We fixed the references to be in parenthesis (\citep instead of \cite)
2. Reviewer's comment: "In Table 5, there is the exact same result between GPT3-ZS and GPT-3 FS in Fall 2021, is that a typo?"
Response: This typo has been fixed with the correct result.
3. Reviewers comment: "We extended this work to include final exams from Harvard's and Cornell's machine learning classes as well. "Are these questions included or to be included in the future?
Response: We included the Harvard and Cornell final exams. Please see the revised version including a dozen exams spanning 17 topics.

Reviewer bhJL
1. Reviewer's comment: “A lot of details are missing with regards to how the dataset was curated. An illustration/figure to show the method used in this work would be advantageous.”
Response: We have added details explaining the curation of the dataset. Please see the supplementary material for data and code.
2. Reviewers comment: “The recommended models do not consider questions that rely on images for their automated solution.”
Response: Images make up only 27.5% of the questions. Gleaning text from input images is beyond the scope of this work.

---

### Decision · Program_Chairs · 2023-01-20

**Decision:**

Reject

**Justification For Why Not Higher Score:**

After discussion, reviewers felt that the paper needs further work to be accepted. Particularly, expanding the dataset to be larger or cover exams in other subjects would substantially strengthen the paper.

**Justification For Why Not Lower Score:**

n/a

**Metareview: Summary, Strengths And Weaknesses:**

The submission presents a new dataset of machine learning exam questions. The paper initially had a wide divergence in reviewer opinion, reflecting disagreements over the usefulness of an additional dataset. Machine learning exams comprise an interesting and challenging domain for evaluating models. However, the proposed dataset is quite small, which reviewers felt would limit interest. Given the vast number of existing datasets, new datasets need to be compelling to make an impact. It would significantly improve the submission to expand the dataset, perhaps to a wider range of subjects than machine learning. However, the idea has much potential, and I would encourage the authors to continue working in this direction and resubmit.

**Summary Of Ac-Reviewer Meeting:**

During discussion, the consensus was that overall the idea here is great, but the execution needs further work. There were legitimate differences of opinion over the importance of dataset papers, but overall the consensus was that the dataset may be unlikely to be widely used in its current form, but that the authors should be encouraged to expand the dataset and resubmit.